# High level risky sexual behavior among persons living with HIV in the urban setting of the highest HIV prevalent areas in Ethiopia: Implications for interventions

**Fekade Wondemagegn**[1], **Tsegaye Berkessa**[2]*

**1** Terkidi Refugee Camp Coordination Office, Gambella, Ethiopia, **2** Department of Public Health, Faculty of Public Health and Medical Sciences, Mettu University, Mettu, Ethiopia

* tsegayebtola@gmail.com

## Abstract

### Introduction

Data on the sexual behavior among people living with human immunodeficiency virus (PLHIV) dwelling at HIV prevalent setting located at the periphery of Ethiopia is lacking. Therefore, this study was designed to investigate sexual practice of patients following their antiretroviral therapy (ART) service and factors affecting their behavior.

### Materials and methods

A facility based cross-sectional study design was employed to assess risky sexual practice and associated factors among HIV positive adults attending ART clinics in Gambella town, Southwest Ethiopia. Risky sexual practice is defined as a custom of getting in at least one of the following practices such as condom-unprotected sex with any partner, having two or more sexual partners and practicing casual sex in the last three months. A total of 352 randomly selected clients were interviewed by using a structured questionnaire. The multivariable logistic regression model was used to examine the association between covariates and the outcome variable.

### Results

Majority of the study participants were engaged at least in one of the risky sexual practices (79.8% confidence interval (CI): 75.3% - 83.9%). The multivariable analysis found that the odds of risky sexual practice were higher among individuals who use substances ('khat' users (AOR: 3.82, 95%CI:1.30–11.22), smoke cigarette (AOR:4.90, 95%CI:1.19–12.60), consume alcohol (AOR: 2.59, 95%CI:1.28–5.21)); those who never discuss about safe sex with their partner/s (AOR: 2.21, 95%CI:1.16–4.21); those who have been in attachment for longer duration (more than four years) with their partner (AOR: 3.56, 95%CI: 1.32–9.62); and groups who desire to bear children in their future life (AOR: 3.15, 95%CI:1.40–7.04) as compared to their respective comparison groups.

**Data Availability Statement:** All relevant data are within the paper and its Supporting Information files.

**Funding:** The author(s) received no specific funding for this work.

**Competing interests:** The authors have declared that no competing interests exist

## Conclusions

A significant number of participants were engaged at least in one of the risky sexual practices which potentially result in super infection by a new or/and drug resistant viral strain/s, and onward transmission of the virus. Thus, an HIV intervention program which focuses on the identified factors has to be implemented to mitigate risk of unsafe sexual behavior of this population group and move towards ending the HIV/Acquired Immunodeficiency Syndrome (AIDS) epidemic.

## Background

HIV/AIDS remains a great public health concern worldwide. According to a recent report, globally 37.9 million people had living with HIV; from this number, about 23.3 million People were access to ART. In 2018, 1.7 million people were infected by HIV, and 770, 000 people died of AIDS-related illnesses [1, 2]. Even though HIV prevalence is reducing from time to time as global trend shows, still new HIV infections are highest among youths living in sub-Sahara Africa. Hence, about five percent of adults in the region are living with HIV [3].

In case of Ethiopia, there were an estimated 23,000 people were newly infected with HIV, 690,000 people living with HIV and there were 11,000 AIDS-related deaths in 2018 [2]. HIV epidemic in Ethiopia varies according to geographic settings. The prevalence of the disease is seven times higher in urban areas compared to the rural areas of the country. Seven out of the nine regional states and two city administrations have an HIV prevalence above one percent. Looking at HIV prevalence by region, it is highest in Gambella (4.8%), followed by Addis Ababa, Dire Dawa and Harari with 3.4%, 2.5%, and 2.4% respectively [4].

Individual-level risks for HIV acquisition and transmission is the core reason for relapsing of the HIV epidemics [5]. Moreover, adolescents and youth living with HIV are at risk of transmitting on the virus to their sexual partners and unborn child. Furthermore, they are vulnerable to potential re-infection with new HIV strain and more vulnerable to other sexually transmitted infections (STIs) compared to their HIV-negative peers [3].

Controversial findings were reported from different studies conducted on the sexual behaviors of PLHIV on ART. Some of the studies conducted in California [6], Uganda [7] and Cameron [8] have noted that a decrease in rates of risky sexual behavior among this population after initiation of ART. On the other hand, several studies from New York [9], Cote D'ivoire [10], Uganda [11], Nigeria [12], and Northwest Ethiopia [13] have shown that they are more likely to engage in risky sexual practice after initiation of ART.

Risky sexual practice/behavior in PLHIV defined in the literature as engaging in one of the following characteristics such as unprotected sex with any partner [13–17], having multiple sexual partners [14–17], casual sex [16, 17], sex under the influence of alcohol [15–17] and sexual exchange (paying or receiving goods or money for sexual intercourse) [17]. A result from cohort studies still advises safe sexual practice is necessary not only to prevent pregnancy and STIs but also to prevent HIV drug-resistant and super infection [18, 19]. A study from Southern Africa found significant associations between risky sexual behaviors (inconsistent condom use and having multiple sexual partners) and HIV infection [20]. Another study conducted in different African countries showed that a change in risky sexual behaviors can reduce HIV prevalence by up to 20% [21]. Non-use of condom by sero-concordant couples encourages the spread of resistant strains of the virus and occurrence of super infection. Super infection that may occur even while under ART in HIV-1 infection [18] was reported from different Africa

countries, among heterosexual couples [22–24]. Moreover, HIV-1 super infection resulting in a triple infection in an HIV-1 infected patient who continues to practice unsafe sex is also documented in Africa [25, 26]. Deterioration of clinical status among HIV infected individuals has been reported as result of super infection [19, 22]. This underscores the need for continued preventive efforts aimed at ensuring safe sexual practices even among HIV-1 sero-concordant couples [19, 27, 28].

To date, limited studies were conducted on the sexual behavior among PLHIV and most of them are conducted at the central part of Ethiopia [13, 16, 17]. In contrary, data on the sexual behavior among people living with HIV in the highly prevalent marginal areas of the country is lacking. Therefore, this study was designed to investigate sexual practice of patients following their ART service and factors affecting their behavior.

## Materials and methods

### Study setting

The study was conducted among adults (18 years or above) living with HIV/AIDS attending ART clinics in Gambella town, Southwest Ethiopia. Gambella town is the capital city of the Gambella Regional state, which is located being 777 km away from Addis Ababa, Ethiopia. The town is administratively structured into 5 keble's (local administrative structure) with 12,928 households and 59,468 total population. There are three governmental health facilities and 12 private clinics located in the town. Only two of the facilities are delivering ART service and a total of 2, 302 clients are actively attending their treatment follow up at ART clinics during the study period.

### Study design

A facility based cross-sectional study design was employed to determine the magnitude of risky sexual practice and associated factors among HIV positive adults attending ART clinics in Gambella town, Southwest Ethiopia using quantitative data collection method.

### Sample size and sampling procedure

The study populations were all HIV positive adults attending ART from June to July 2019 at ART clinics in Gambella town. The sample size was calculated by using single proportion formula, the magnitude of risky sexual practice was 38% from previously conducted research in Gondar town, Northwest Ethiopia [13], with the marginal error tolerated (d) to be 5%, and 95% confidence level giving a sample size of 362. With regards to the current study, since the source population was less than 10,000, finite population corrections formula was used to get a sample size of 313. Finally, by adding the non response rate of 15%, the total final sample size was 360. Procedurally, the sample from each health facility was proportionally allocated, and then every sixth was selected for interview by using systematic random sampling technique.

### Data collection and measurement

A structured questionnaire which was adopted with modifications [5, 16] and pretested was used for data collection. The pretest was done on 30 participants attending ART clinics at neighbor town prior to the actual data collection and the questionnaire was modified when necessary. The questionnaire was first prepared in English and translated into the Amharic language, and then retranslated into English by language experts to check the consistency. The main outcome variable for the study was risky sexual practice, defined as engaged at least in one of the following practices such as condom-unprotected sex with any sexual partner, having two or more sexual partners and casual sex in the last three months prior to the date of data

collection [16, 17]. The independent variables were socio-demographic characteristics which include sex, age, ethnicity, educational status, religion, marital status, occupation, monthly income, and family size; behavioral and social factors like substance use (alcohol, cigaratte and 'khat'), desire of child, attendance of HIV prevention discussion, safer sex behavior skill training, stigma, length of followup and CD4 count; sexual practice and partner related factors like discussion about safe sex, partner HIV status and disclosure status; medical related factors such as duration of diagnosis of HIV and CD4 count. Steady partner was a partner with whom a respondent had regular sexual relationship and perceived by them as spouse or regular boy/girlfriend [17]. Casual partner means individuals with whom they had sexual intercourse once or a few times other than regular steady partners (spouse/boy/girlfriend) with or without payment [17]. 'Khat' use was defined consumption of 'khat' (chewing) during the last month prior to data collection date [29]. Alcohol consumption was defined as consuming >4 drinks in a day (or >14 drinks/week) for men and >3/day (or >7/week) for women [30]. Cigaratte smokers was defined smokers reporting one cigarette per day or an average of at least seven cigarettes per week [29].

## Data processing and analysis

Data were entered using the Epidata 3.1 software, and then exported to the STATA version 15 (College Station, TX, USA) for descriptive, bivariable and multivariable analysis. The model was fitted by Hosmer and Lemeshow's goodness-of-fit [31]. All the variables with P-value < 0.25 with risky sexual practice in univariable analysis were fitted to the final multivariable logistic regression model. Multi-Collinearity was checked using Variance Inflation Factor (VIF); values < 10 were included in the model. In the multivariable analysis, a value of P<0.05 was considered as statistically significant association. Odds ratio (OR) with 95% confidence level was used to show the strength of association between dependent and independent variables.

## Ethics statement

Ethical clearance was granted by the Mettu University Institutional Review Board (IRB). Moreover, a support letter to conduct the study was obtained from Gambella regional health office. Respondents were also informed about the purpose, procedure, possible risks and benefits of participating in the study and the confidentiality of information they provide. Thus, participation in the study was voluntary, and patients had the right to stop the interview at any time. In nutshell, data were collected after informed verbal consent was obtained from each participant and name or other personal identification of the participants of the study were made anonymous. data were collected in the questionnaire.

## Results

### Socio-demographic characteristics

A total of 352 participants were interviewed in this study with a response rate of 97.8%, out of which, 102 (29%) of them were Aynwa by ethnicity. The mean age of the respondents was 34.1 years (SD±9.8 years). About half of them were females and two third were married. On the top of these more than half of the respondents had a monthly average income below three thousand Ethiopian Birr (Table 1).

### Behavioral, social and medical characteristics

In this study, one fourth of participants often use 'khat', 140 (39.8%) and 98 (27.8%) of them consume alcohol and smoke cigarette respectively. Regarding fertility desire 291 (82.7%) of

**Table 1. Socio-demographic characteristics of adults living with HIV attending ART clinics in Gambella town, Southwest Ethiopia.**

| Characteristics | | Frequency | Percentage |
|---|---|---|---|
| Age (in years) | 18–29 | 133 | 37.8 |
| | 30–39 | 134 | 38.1 |
| | ≥ 40 | 85 | 24.1 |
| Sex | Female | 180 | 51.1 |
| | Male | 172 | 48.9 |
| Ethnic group | Anywa | 102 | 29.0 |
| | Amhara | 79 | 22.4 |
| | Nuer | 66 | 18.8 |
| | Oromo | 65 | 18.5 |
| | Other | 40 | 11.4 |
| Marital status | Married | 230 | 65.3 |
| | Single | 51 | 14.5 |
| | Divorced | 46 | 13.1 |
| | Widowed | 25 | 7.1 |
| Educational level | Primary education | 90 | 25.6 |
| | Unable to read and write | 73 | 20.7 |
| | Technical/vocational collage | 73 | 20.7 |
| | Secondary education | 56 | 15.9 |
| | Informal education | 32 | 9.1 |
| | Degree and above | 28 | 8.0 |
| Occupation | Self-business | 107 | 30.4 |
| | Government employee | 95 | 27.0 |
| | Daily labor | 65 | 18.5 |
| | House wife | 57 | 16.2 |
| | Other | 28 | 8.0 |
| Religion | Protestant | 158 | 44.9 |
| | Orthodox | 120 | 34.1 |
| | Catholic | 44 | 12.5 |
| | Muslim | 28 | 8.0 |
| | Other | 2 | 0.6 |
| Monthly average income (ETB)* | ≤1500** | 55 | 15.6 |
| | 1501–3000 | 131 | 37.2 |
| | > 3000 | 166 | 47.2 |
| Family size | ≤3 | 144 | 40.9 |
| | 4–7 | 156 | 44.3 |
| | > = 8 | 52 | 14.8 |

*Ethiopian Birr

** extreme poverty (less than 1.90 $ perday)

respondents desire to bear children in the future, of which majority of the respondents, 254 (72.2%) want to have two and more children. Among the females who participated in this study, about two for every seven females had a history of pregnancy in the last twelve months and 22 (44.9%) of them had intended pregnancy.

Concerning health related services; about one fourth of them have attended support group discussion on the safe sex, 29 (8.2%) of them on their part participated in skill building training on safer sex behaviors. Regarding stigma, 35 (9.9%) and 12 (3.4%) of them experienced

perceived and enacted stigma respectively. About forty-five percent of respondents have already started ART medication two years ago before data collection date. The majority of participants, 343 (97.4%) had CD4 count >350 cells/mm3 (Table 2).

## Magnitude of risky sexual practices and partner related characteristics

Majority of the respondents had engaged in at least one of the risky sexual practices. Ninety-six (27.3%) had multiple partners, 66 (18.8%) with a casual partner and 58 (16.5%) with both steady and casual partners. Regarding condom use; 274 (77.8%) of them reported indicating as they inconsistently used or never used at all in all their sexual intercourse during past three months preceding the date of data collection. Different reasons were mentioned by the study participants for not using at all or inconsistently using of condom (Fig 1).

About forty-one percent of the PLHIV was reported that their sexual partner HIV sero-status was negative or unknown whereas majority, 262 (74.4%) of their sexual partners were aware about their status. Most of the participants stated that they were staying with their current partner for more than one year, and 160 (45.5%) of them were discussed about safe sex with their partner/s.

## Factors associated with risky sexual practice

The multivariable analysis found that the odds of risky sexual practice were higher among individuals who use substances ('khat' users (AOR: 3.82, 95%CI:1.30–11.22), smoke cigarette (AOR:4.90, 95%CI:1.19–12.60), consume alcohol (AOR: 2.59, 95%CI:1.28–5.21)); those who never discuss about safe sex with their partner/s (AOR: 2.21, 95%CI:1.16–4.21); those who have been in attachment for longer duration (more than four years) with their partner (AOR: 3.56, 95%CI: 1.32–9.62); and groups who desire to bear children in their future life (AOR: 3.15, 95%CI:1.40–7.04) as compared to their respective comparison groups (Table 3).

## Discussions

The current study explored the sexual risk behaviors among HIV-positive patients taking ART in Gambella town, Southwest Ethiopia. The study depicted that the high rates of sexual risk behavior among HIV-positive individuals on the ART have implications for the risk of contracting and /or transmitting the virus in the study area. In the curent study, the researches found that a majority of patients (77.8%) experienced inconsistent use of condom or never used it at all. From those who have negative or unknown HIV sero-status partner/s, about 77% of them had one or more sexual encounter(s) without using a condom in the last three months prior to data collection period. This highlights the dangers of continued HIV transmission despite the increasing ART rollout. Therefore, these findings call for rising awareness and motivation of using condom among HIV-positive patients. A substantial number of HIV positive clients, which accounts for (27.3%) had practiced a sexual intercourse with at least two partners. Furthermore, the findings of this study revealed that sexual practices of this vulnerable population, and underscore ways of intervening problems related to unsafe sexual practice.

This study shows that 79.8% (CI: 75.3% - 83.9%) of the respondents had at least one risky sexual practice within three months prior to the study. This finding is consistent with the result (81%) reported from Uganda (15) but it is higher than study reported (70.6%) from Southeast Nigeria [32]. It is also higher than previously reported results from Addis Ababa and Gondar, other parts of the country; where the magnitude of risky sex was 36.9%, and 38% respectively [13, 17]. The possible reason for the difference might be due to socio-demographic and geographical variation between the previous two towns compared to the current study setting. Other possible reasons could be variation in the operational definition of the risky sexual

**Table 2. Behavioral, social and medical characteristics of adults living with HIV attending ART clinics in Gambella town, Southwest Ethiopia.**

| Characteristics | Frequency | Percentage |
|---|---|---|
| 'Khat' use | | |
| Yes | 88 | 25.0 |
| No | 264 | 75.0 |
| Cigarette smoking | | |
| Yes | 98 | 27.8 |
| No | 254 | 72.2 |
| Alcohol consumption | | |
| Yes | 140 | 39.8 |
| No | 212 | 60.2 |
| Other substance use* | | |
| Yes | 69 | 19.6 |
| No | 283 | 80.4 |
| Desire of children in the future | | |
| Yes | 291 | 82.7 |
| No | 61 | 17.3 |
| Number of desired children | | |
| 1 | 37 | 12.7 |
| 2 | 201 | 69.1 |
| ≥ 3 | 53 | 18.2 |
| History of pregnancy in the past 12 months (females) | | |
| Yes | 49 | 27.2 |
| No | 131 | 72.8 |
| Intended pregnancy | | |
| Yes | 22 | 44.9 |
| No | 27 | 55.1 |
| Caused a pregnancy in the past 12 months (males) | | |
| Yes | 22 | 12.8 |
| No | 150 | 87.2 |
| Attending support group discussion on HIV prevention | | |
| Yes | 91 | 25.9 |
| No | 261 | 74.1 |
| Receiving any skill training on safer sex behaviors | | |
| Yes | 29 | 8.2 |
| No | 323 | 91.8 |
| Perceive stigma | | |
| Yes | 35 | 9.9 |
| No | 317 | 90.1 |
| Enact stigma | | |
| Yes | 12 | 3.4 |
| No | 340 | 96.6 |
| Length of follow up care (in months) | | |
| ≤ 12 | 29 | 8.2 |
| 13–48 | 167 | 47.4 |
| ≥ 49 | 156 | 44.3 |
| Current CD4 count | | |
| ≤350 | 9 | 2.6 |

(*Continued*)

**Table 2.** (Continued)

| Characteristics | Frequency | Percentage |
|---|---|---|
| >350 | 343 | 97.4 |

*Substance indicates Shisha/hashish

practice as the study conducted in Addis Ababa used only a single character of risky sexual practice, condom-unprotected sex with any partner. High risky sexual practice in this study highlights behavioral interventions that can reduce unsafe sexual practice among PLHIV should be reinforced.

In the current study, substance use (alcohol, 'khat' and tobacco) was associated with risky sexual behavior. We found out that alcohol consumption was found to be significantly associated with their high-risk sexual behavior. A similar finding was also reported from a cohort study in Switzerland [33], Northern India [34], Togo [35], Southwestern Uganda [36] and Kenya [37]. A meta-analysis study has also identified alcohol as correlate of unprotected sexual behavior [38]. The correlation between alcohol consumption and risky sexual practice might be due to decreased self-consciousness and impaired judgments after alcohol intake which may in turn increase risky sexual practice. Odds of risky sexual practice increased by four and five folds among 'Khat' chewing and cigarette smoking groups respectively compared to non-users. The association between these two substances and risky sexual behavior was reported in another population. A study conducted in Malaysia reported indicating as there significant correlation between smoking and sexual activity [39]. A community based study in Ethiopia

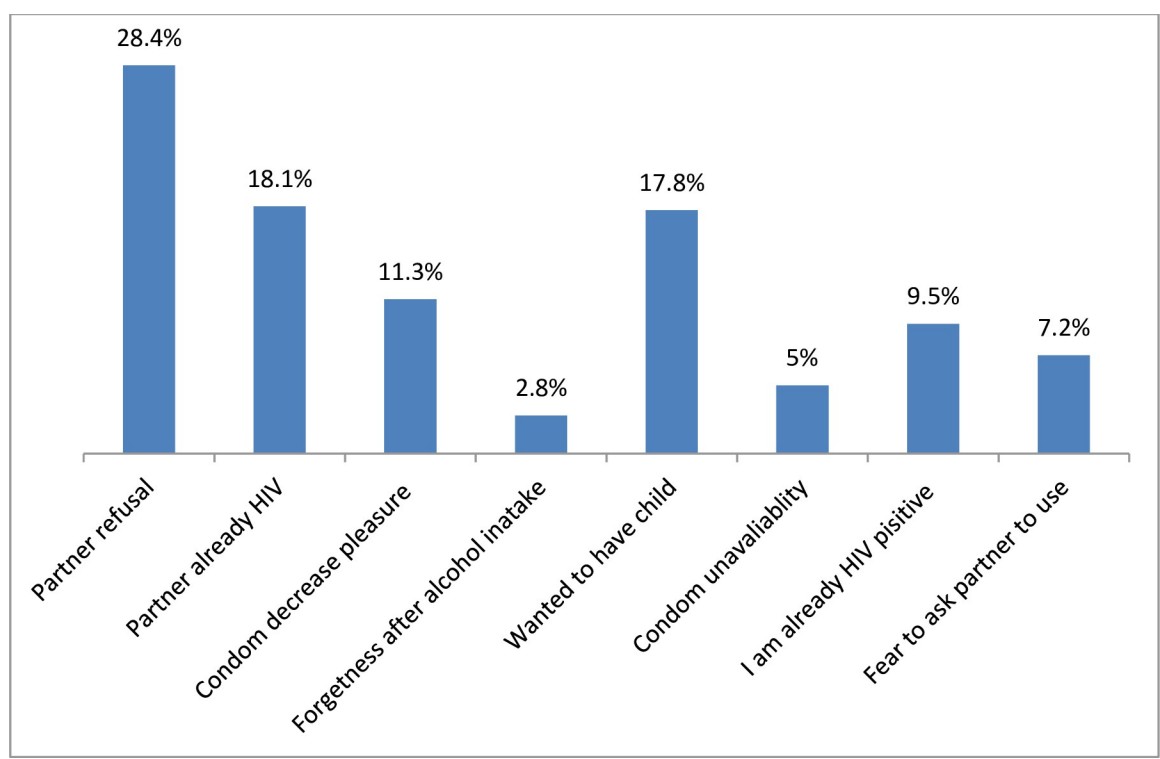

**Fig 1. Reasons for not using at all or inconsistently using of condom among adults living with HIV attending ART clinics in Gambella town, Southwest, Ethiopia, 2019.**

**Table 3. Bivariable and multivariable logistic regression analysis of factors associated with risky sexual practice among adults living with HIV attending ART clinics in Gambella town, Southwest Ethiopia.**

| Characteristics | Risky sexual practice | | COR (95%CI) | P-value | AOR (95%CI) | p-value |
|---|---|---|---|---|---|---|
| | No | Yes | | | | |
| Marital status | | | | | | |
| Married | 36 | 194 | 1 | | 1 | |
| Single | 13 | 38 | 0.54(0.26–1.12) | 0.097 | 0.84(0.30–2.34) | 0.735 |
| Others[a] | 22 | 49 | 0.41(0.22–0.77) | 0.005 | 0.72(0.30–1.72) | 0.465 |
| Average monthly income (in ETB)[b] | | | | | | |
| <1500 | 18 | 37 | 1 | | 1 | |
| 1500–300 | 25 | 106 | 2.06(1.01–4.20) | 0.046 | 1.52(0.65–3.52) | 0.322 |
| >300 | 28 | 138 | 2.40(1.20–4.80) | 0.014 | 1.23(0.53–2.88) | 0.626 |
| Discussion about safe sex with partner/s | | | | | | |
| Yes | 41 | 119 | 1 | | 1 | |
| No | 30 | 162 | 1.86(1.10–3.15) | 0.021 | 2.21(1.16–4.21) | 0.016* |
| Length of stay with current partner/s (in months) | | | | | | |
| ≤12 | 22 | 48 | 1 | | 1 | |
| 13–48 | 21 | 67 | 1.46(0.72–2.95) | 0.290 | 1.78(0.72–4.40) | 0.214 |
| ≥49 | 28 | 166 | 2.72(1.43–5.17) | 0.002 | 3.56(1.32–9.62) | 0.012* |
| 'Khat' use | | | | | | |
| No | 66 | 198 | 1 | | 1 | |
| Yes | 5 | 83 | 5.53(2.15–14.23) | 0.000 | 3.82(1.30–11.22) | 0.015* |
| Cigarette smoking | | | | | | |
| No | 65 | 189 | 1 | | 1 | |
| Yes | 6 | 92 | 5.27(2.20–12.62) | 0.000 | 4.90(1.91–12.60) | 0.001** |
| Shisha/hashish use | | | | | | |
| No | 64 | 219 | 1 | | 1 | |
| Yes | 7 | 62 | 2.59(1.13–5.93) | 0.025 | 1.35(0.53–3.48) | 0.529 |
| Alcohol consumption | | | | | | |
| No | 55 | 157 | 1 | | 1 | |
| Yes | 16 | 124 | 2.71(1.48–4.97) | 0.001 | 2.59(1.28–5.21) | 0.008* |
| Desire of children in the future | | | | | | |
| No | 21 | 40 | 1 | | 1 | |
| Yes | 50 | 241 | 2.53(1.38–4.66) | 0.003 | 3.15(1.40–7.04) | 0.005* |
| Length of follow up on ART | | | | | | |
| ≤12 | 11 | 18 | 1 | | 1 | |
| 13–48 | 31 | 136 | 2.68(1.15–6.24) | 0.022 | 1.69(0.56–5.06) | 0.349 |
| ≥49 | 29 | 127 | 2.68(1.14–6.27) | 0.023 | 1.88(0.63–5.63) | 0.256 |
| Receiving any training / skill building on safe sex | | | | | | |
| Yes | 9 | 20 | | | 1 | |
| No | 62 | 265 | 1.89(0.82–4.36) | 0.133 | 1.92(0.68–5.47) | 0.219 |
| Attending support group discussion on HIV | | | | | | |
| Yes | 13 | 78 | | | 1 | |
| No | 58 | 203 | 0.58(0.30–1.12) | 0.107 | 0.71(0.32–1.55) | 0.390 |

Note:1 = reference

[a]widowed and divorced marital status

[b]Ethiopian birr

*p-value< 0.05

**P<0.01

also showed that 'Khat' consumption is associated with HIV risk behavior [40, 41]. In Ethiopia, there are habitual practices that smoking and alcohol consumption after 'Khat' chewing practice [41, 42]. Using a substance in combination reduces inhibitions and increases the vulnerability of risky sex [41]. In addition to risky sexual practices, substance use is significantly contributed to poor ART adherence and poor HIV medical outcomes [43–45]. This information is critical for the development of policy and practice for HIV/AIDS care including the prioritization and planning of effective substance use screening tools and intervention methods.

A study from USA indicated that one of the most important methods to prevent HIV transmission is interpersonal communication which results in reaching on the consensus through free and frank discussion about safer sexual behavior [46]. In this study, those who did not discuss safe sex with their partner/s were 2.21 times more engaged in risky sexual practice than their counterparts. This finding is similar with previous studies conducted in other parts of the country [13, 47]. This might be due to the fact that discussion safe sex may avoid engaging in unprotected sexual acts in both sexes. Hence, avoiding open discussion on safe sex may potentially make the partners to engage in risky sex.

Similar to previous result from Eastern part of the country, this research has identified that those who have stated together for more than 4 years with a partner were more likely to be engaged in the risky sexual practice than those who stay less than one year with their partner [47]. The possible reason was that they stayed together for longer duration, and then they trust each other as a result, they might tend to be engaged in risky sexual practice. Similar to other studies [48, 49], desire to have a child is significantly associated with risky sexual practice. This might be due to social and cultural contexts put pressure on couples to bear children as a demand that couples have to fulfill in the marriage.

In this study, more than 80% of the participants have future fertility desire. Except for the use of screened fresh sperm from HIV sero-negative donors (when a woman's male partner is HIV-infected) and adoption no conception methods are a complete risk-free of HIV transmission. However, some risk reduction methods have been used in the developed world for safer conception [50]. In resource-limited setting like Ethiopia, promotion of safer conception counseling, strong adherence to ART to reduce infectiousness of PLHIV and preventing the spread of HIV from mother to child (PMTCT) services is yet the possible ways to reduce the risk of infection.

A number of potential limitations may affect the findings of this study. First, sample size was only calculated for the prevalence of risky sexual practice and other associated factors were not considered. Secondly, sensitive of the subject may result in social desirability bias. Lastly, due to the limitation of cross-sectional study design, it is impossible to establish causal relationship and further longitudinal research is warranted to investigate the effects of the factors on the sexual behaviors among PLHIV. However, as the objective of the study was to determine the magnitude of risky sexual behavior by using a comprehensive definition and potential factors associated with it, the findings could serve as an important input to inform proper target of HIV intervention program in this population group towards ending AIDS.

## Conclusions

In conclusion, the magnitude of risky sexual practice defined as engaged at least in one of the following practices: having two or more partners, causal sex and condom-unprotected sex with any partner in the last three months among HIV positive individuals who attended ART clinics in Gambella town was very high. This indicates that a considerable number of clients were potentially exposed to and/or causing super infection by a new and/or drug resistant viral strain/s, and also can infect the unborn child and their HIV sero-negative sexual partner/s.

The study identified that substance use (alcohol, 'khat' and cigarette), lack of discussion about safe sex among sexual partners', desire to have a child in the future, and staying together for long duration (more than four years) with a partner were important predictors of the risky sexual practice. Thus, an HIV intervention program which focuses on the identified factors has to be implemented to mitigate risk of unsafe sexual behavior in this population group and move towards ending of the HIV/AIDS epidemic.

## Supporting information

**S1 File. Consent form and questionnaire (English version).**
(PDF)

**S2 File. Consent form and questionnaire (Amharic version).**
(PDF)

## Acknowledgments

Our sincere thanks go to the Mettu University for the ethical approval of this reseach. We are grateful to the Gambella regional health biro, Gambella referral hospital and Gambella health center for their cooperation and for giving us all the invaluable information we requested. Finally, we offer our gratitude to the study participants, as well as the supervisors, data collectors and all others who made this study possible.

## Author Contributions

**Conceptualization:** Fekade Wondemagegn, Tsegaye Berkessa.

**Data curation:** Fekade Wondemagegn, Tsegaye Berkessa.

**Formal analysis:** Tsegaye Berkessa.

**Methodology:** Fekade Wondemagegn, Tsegaye Berkessa.

**Software:** Tsegaye Berkessa.

**Validation:** Tsegaye Berkessa.

**Writing – original draft:** Fekade Wondemagegn, Tsegaye Berkessa.

**Writing – review & editing:** Fekade Wondemagegn, Tsegaye Berkessa.

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
