## [Decision Letter · Decision Letter 0]

21 Jul 2020

PONE-D-20-13455

High level of risky sexual behavior among persons living with HIV in the urban setting of the highest HIV prevalence in Ethiopia: Implications for HIV interventions

PLOS ONE

Dear Dr. Tola,

Thank you for submitting your manuscript to PLOS ONE. After careful consideration, we feel that it has merit but does not fully meet PLOS ONE’s publication criteria as it currently stands. Therefore, we invite you to submit a revised version of the manuscript that addresses the points raised during the review process.

The manuscript has many typos and grammatic issues and must be copy-edited before the upcoming submission.More than 80% of the respondents have future fertility desire. What do you recommend for balancing risky sexual behaviour and fertility desire in PLWHA including those represented in the study? Please discuss the matter further.Can you please provide a brief review (a paragraph or so) on how the existing literature define risky sexual behaviour in PLWHA and justify the definition used in the study?The operational definitions for many of the key variables including Khat chewing, alcohol use, smoking have not been provided. Please also make sure that the definitions on substance use take dose and frequency of use into consideration. Terms like casual partner, steady partner must also be defined.The variables of the study are not clearly provided in the paper. As it stands, it is not possible to identify those variables that did not make it to the multivariable model. In the methods section please add a section that exhaustively describes the independent variables of the study (along with their operational definitions).Please comment on the adequacy of the sample size for identifying predictors of risky sexual behaviour.The rationale for considering the variables (HIV positive child living with family, family size) as predictors of risky sexual behaviour is not clear. Please justify or remove them from the analysis.Marital status: Separated and divorced levels should be merged. As it stands the frequencies are too small for reasonable analysis.Table 3: The variable “Length of stay with current partner/s” is confusing. What would happen if the respondent had no partner in the reference period? How did you calculate the duration among those study participants who had multiple partners?I don’t see any report on the association between clinical variables (e.g. HIV disease staging, viral load/CD4 count) and risk sexual behaviour? Where they considered as independent predictors in the study?

We look forward to receiving your revised manuscript.

Kind regards,

Samson Gebremedhin, PhD

Academic Editor

PLOS ONE

Journal Requirements:

https://journals.plos.org/plosone/article?id=10.1371%2Fjournal.pone.0174267

https://bmcpublichealth.biomedcentral.com/articles/10.1186/1471-2458-11-422

https://www.ncbi.nlm.nih.gov/pubmed/27190412

In your revision ensure you cite all your sources (including your own works), and quote or rephrase any duplicated text outside the methods section. Further consideration is dependent on these concerns being addressed.

3. Please address the following:

- Please include additional information regarding the survey or questionnaire used in the study and ensure that you have provided sufficient details that others could replicate the analyses. For instance, if you developed a questionnaire as part of this study and it is not under a copyright more restrictive than CC-BY, please include a copy, in both the original language and English, as Supporting Information. In addition, please provide further details concerning the pretesting of this tool, including the number of participants and where they were recruited from.

- Please provide additional details regarding participant consent. In the ethics statement in the Methods and online submission information, please ensure that you have specified how verbal consent was documented and witnessed.

Additional Editor Comments (section-by-section comments):

Abstract

The background sub-section of the abstract is difficult to follow. Please revise and make it focused on the knowledge gap that lead to this undertaking.In the methods section please provide the operational definition for “risky sexual practice”The sentence “Finally, variables with p<0.05 was considered statistically significant” is less relevant in the abstract section.Write abbreviations in expanded form at first use, both in the abstract and main body of the manuscurpt. 

Methods

Line 105: confidence interval >> confidence levelPlease provide a separate section on “data collection and measurement” and integrate the “operational definitions” sub-section with this section.Please check for the absence of multicolinarity among the independent variables of the study

Discussion

The discussion looks more of a literature review. Please discuss the interpretation, implication and methodological limitations of the study.

Conclusion

The conclusion section must be condensed into one short paragraph. 

Other comments

Multivariate >> multivariableBivariate >> bivariable

Reviewers' comments:

Reviewer's Responses to Questions

**Comments to the Author**

1. Is the manuscript technically sound, and do the data support the conclusions?

Reviewer #1: Partly

Reviewer #2: Yes

2. Has the statistical analysis been performed appropriately and rigorously? 

Reviewer #1: Yes

Reviewer #2: Yes

3. Have the authors made all data underlying the findings in their manuscript fully available?

Reviewer #1: No

Reviewer #2: Yes

4. Is the manuscript presented in an intelligible fashion and written in standard English?

Reviewer #1: No

Reviewer #2: Yes

5. Review Comments to the Author

Reviewer #1: There is major flaw in the operational definition of respondents’ sexual behavior/practice which significantly affects the study finding, discussion and conclusion sections alike. Please see the attached detail review feedback.

Reviewer #2: The paper “High level of risky sexual behavior among persons living with HIV in the urban setting of the highest HIV prevalence in Ethiopia: Implications for HIV interventions” is very interesting. It is important to note that the authors were careful in writing and that the text has great potential for publication, in addition to dealing with a relevant theme in the literature. Some comments, however, are necessary to improve the quality of the manuscript.

6. PLOS authors have the option to publish the peer review history of their article (what does this mean?). If published, this will include your full peer review and any attached files.

Reviewer #1: No

Reviewer #2: No

---

## [Author Response · Author response to Decision Letter 0]

4 Sep 2020

Dear Editor:

I appreciate the comments and suggestions of the academic editor and reviewers; their comments and suggestions were constructive and have improved our manuscript substantially. Accordingly, I have incorporated almost all of the comments and suggestions given by the academic editor and reviewers in the manuscript. Furthermore, I have responded to each of the points raised by the reviewers as follows: 

Sincerely,

Tsegaye Berkessa (Corresponding author)

Academic Editor:

• The manuscript has many typos and grammatic issues and must be copy-edited before the upcoming submission.

Response: The comment was accepted and the manuscript was edited.

• More than 80% of the respondents have future fertility desire. What do you recommend for balancing risky sexual behaviour and fertility desire in PLWHA including those represented in the study? Please discuss the matter further.

Response: This comment has been well taken and corrected accordingly.

• Can you please provide a brief review (a paragraph or so) on how the existing literature define risky sexual behaviour in PLWHA and justify the definition used in the study?

Response: The comment was accepted and correction was made accordingly

• The operational definitions for many of the key variables including Khat chewing, alcohol use, smoking have not been provided. Please also make sure that the definitions on substance use take dose and frequency of use into consideration. Terms like casual partner, steady partner must also be defined.

Response: The comment was accepted and definition of terminologies was done.

• The variables of the study are not clearly provided in the paper. As it stands, it is not possible to identify those variables that did not make it to the multivariable model. In the methods section please add a section that exhaustively describes the independent variables of the study (along with their operational definitions).

Response: Correction was made accordingly

• Please comment on the adequacy of the sample size for identifying predictors of risky sexual behavior

Response: Our sample size was slightly limited compared to the number of independent variables. However, we excluded less important variables and less important variable categories was also merged. Finally, limited candidate predictors was examined to have the lower limit of events per variable (EPV) for developing the prediction model. 

• The rationale for considering the variables (HIV positive child living with family, family size) as predictors of risky sexual behaviour is not clear. Please justify or remove them from the analysis.

Response: The comment has been well taken and removed from the analysis.

• Marital status: Separated and divorced levels should be merged. As it stands the frequencies are too small for reasonable analysis.

Response: This is definitely correct and we have modified accordingly.

• Table 3: The variable “Length of stay with current partner/s” is confusing. What would happen if the respondent had no partner in the reference period? How did you calculate the duration among those study participants who had multiple partners?

Response: In our study, all people living with HIV was reported they had at least one sexual partner in the last three months. Study participants were asked whether they had multiple sexual partners in the last three months. Similar to other studies for those who had a single partner on the date of data collection and separated from the other partner/s in the last three months the current partner duration was taken and for the others who had multiple partners on the date of data collection the longest stay was considered. 

• I don’t see any report on the association between clinical variables (e.g. HIV disease staging, viral load/CD4 count) and risk sexual behaviour? Where they considered as independent predictors in the study?

Response: The comment was accepted and CD4 count was included in the model. 

Abstract

• The background sub-section of the abstract is difficult to follow. Please revise and make it focused on the knowledge gap that lead to this undertaking.

• In the methods section please provide the operational definition for “risky sexual practice”

• The sentence “Finally, variables with p<0.05 was considered statistically significant” is less relevant in the abstract section.

• Write abbreviations in expanded form at first use, both in the abstract and main body of the manuscurpt. 

Response: The comment was accepted and abstract revision was done.

Methods

• Line 105: confidence interval >> confidence level

• Please provide a separate section on “data collection and measurement” and integrate the “operational definitions” sub-section with this section.

• Please check for the absence of multicolinarity among the independent variables of the study

Response: The methods part was revised accordingly.

Discussion

• The discussion looks more of a literature review. Please discuss the interpretation, implication and methodological limitations of the study.

Response: Correction was made accordingly

Conclusion

• The conclusion section must be condensed into one short paragraph. 

Response: The comment was accepted and correction was made accordingly

Other comments

• Multivariate >> multivariable

• Bivariate >> bivariable

Response: Revision was done accordingly

 

Reviewer 1:

• The scientific writing and English language needs editorial work.

Response: The comment was accepted and edition was made accordingly

• There is major flaw in the operational definition of respondents’ sexual behavior/practice which significantly affects the study finding, discussion and conclusion sections alike.

Response: The operational definition risky sexual practice in our study was used in several previously published reputable literature. Explained in detail in the specific feedbacks. 

Specific feedbacks:

• The HIV prevalence in Ethiopia is technically “concentrated” (less than 1%) according to UNAIDS definition while the average in Africa is “generalized” 5% as indicated in the background of the manuscript (Line 50). Nonetheless, the authors characterized Ethiopia’s HIV epidemic as one of the highest in Africa which is not correct (see the Abstract and line 51 without any reference).

Response: This comment has been well taken and corrected accordingly

• This study has critical deficiency/limitation on the operational definition of “risky sexual behavior/practice” (specifically the consistent condom use …line 116) which I believe leads to remarkable challenges/gaps across the result, discussion and conclusion sections. Respondents within marital union shouldn’t necessary be categorized as “risky sexual practice” if they aren’t using condom within the union unless they are sero-discordant. However, the operational definition doesn’t qualify so. This resulted in a very higher proportion of respondents as a risky sexual practices (80%) which is not comparable with some local studies done in country (39% and 38% in Addis Ababa and Gondar respectively……line 205-206). This issue affects finding in the result (including the regression model), discussion, and ultimately the conclusion. Table-1 (Line 143) demonstrates the importance of this issue as 65% of the respondents are married. Response: The definition of “risky sexual practice” is varied from one study to another. Risky sexual practice/behavior in PLHIV defined in the literature as engaging in one of the following characteristics such as condom-unprotected sex with any sexual partner, having multiple sexual partners, casual sex, sex under the influence of alcohol and sexual exchange within the last three months (cited in the revised introduction section). Regarding condom-unprotected sex, a few studies consider as risky sexual practice only if unprotected sex with negative or unknown HIV status partner. This might be the concern of these studies were only whether the people living with HIV are adding a new infection to the pool or not. However, most of the literature was used unprotected sex with any partner including spouse/cohabiting partner (regardless of their HIV sero status) (cited in the revised manuscript). The main reason is non use of condom by sero concordant couples encourages the spread of resistant strains of the virus and occurrence of super infection. Super infection/re infection that cause detrimental clinical effects can occur even while under ART in HIV-1 infection and reported from different Africa countries among heterosexual couples. As result, still several longitudinal studies underscores the need for continued preventive efforts aimed at ensuring safe sexual practices even among HIV-1 sero-concordant couples (well cited in the revised introduction section). In addition, even though HIV-seropositive individuals deserve full reproductive rights, no conception methods are 100% risk-free of HIV transmission, other than the use of screened fresh sperm from HIV-seronegative donors (when a woman’s male partner is HIV-infected) and adoption. Hence, it is difficult to take unprotected sex among regular PLHIV partner as a safe practice. All most all of previously locally conducted studies (in Addis Aba (Dessie et al., 2011, Tadesse et al., 2019), in Gondor (Molla et al., 2017) and other several studies conducted in Africa used condom-unprotected sex with any partner as one character of unsafe sex practice (risky practice) (cited in the revised manuscript). The variation between the result (36.9% and 38% in Addis Ababa and Gondar) is not due to the difference in contextual definition of condom use among regular partner (marital union) because we used similar definition in this regard. A study conducted at Gondar defined “risky sexual practice as having one or more of the following practices during the past three months prior to date of data collection: having multiple sexual partners, casual sex, sex without or inconsistent use of condom even with regular partner, sex with the influence of substance like alcohol”. We used similar definition except for sex with the influence of substance like alcohol. If you look at the result from this study, from 38% more than 80% of them were classified as risky due to condom-unprotected sex. More than half (51.7%) of them were married and even being married was identified as one of the predictor of risky sexual practice (by 6 fold). A study conducted by Dessie and his colleagues at Addis Ababa city was defined “risky sexual practices as condom-unprotected sex with either HIV negative, HIV positive or unknown sero status partner”. In this study, about 64% of participants were married and from 36.9% risky sexual practice, majority (77.0%) of them were due to unprotected sex with spouse/regular partner (it includes marital union). Generally, higher risky sexual practice in our study compared to previous local studies (Gondar & Addis Ababa) is not due to the limitation/deficiency of our operational definition and ‘this definition’ was used by several previously published reputable literature. 

• Line 157-158= It would have been good to use Viral Load levels instead of CD4 since the latter has been phased out as a treatment outcome monitoring platform for PLHIV on treatment unless this data collection was done before the change of protocol. 

Response: Response: Yes exactly, data collection was done before implementation of the new protocol. 

• Line 163= Says “four out of five respondent” which is 80% of respondents engaged in risky sexual practice, is this to say at least one of the operationally defined risky practices? If so it has to be clearly articulated so across the manuscript.

 Response: This is definitely correct and we have modified accordingly 

• Line 164-168= All risky sexual practices are less than 27% except consistent condom use (77.8%) which has exaggerated the overall finding as mentioned above.

Response: Looking at each risky sexual character multiple sexual partner is 27.3%, Casual sex is 35% (124/352 ⁓ 66 casual + 58 both casual and steady partner) and condom unprotected sex is 77.8%. However, over all risky sexual practice in the absence of condom-unprotected sex is still 43.2% (at least one risky practice; casual or had 2 or more partners). Moreover, in this study if we exclude condom-unprotected sex among HIV sere-concordant regular partners (within marital union) still the overall risky sexual practice is high (58.5%). 

• Line 169=58.8% respondents’ partners’ are sero-positive, this statement is misleading unless the term “partner” is defined well and the denominator is mentioned. As it stands now, it mean more than 40% of PLHIV on treatment are in a sero-discordant relationship which is unrealistically high. The same clarity is required for partner notification on the same line (169).

Response: Yes, this line lacks clarity and can mislead and correction was made by making it clear. The left 40.2 % is not necessarily sero-discordant because it include both sero negative and unknown sero status (18.2% seronegative and 23% unknown sero status). 

• Line 186= Table-3 Bivariate and Multivariate LR analysis, has core deficiency and need to be remodeled once the operational definition issue is addressed. 

Response: We used similar operational definition with other several literature including the local studies as we justified in the other comment. 

• Line 190-193= The remarks based on the finding is unrealistic

 Response: The comment was accepted and revision was made accordingly

• Line 193-195= There is inconsistency between the finding and subsequent remark 

Response: The comment was accepted and revised accordingly

• Line 201= see comment (feedback) on Line 163

Response: The comment was accepted and correction was made accordingly

• Line 205-206= The discussion pointed out important issue about the exaggerated risk level by comparing it with other studies (Addis Ababa and Gondar) and one of the authors’ possible justification is the way risk sexual practice is defined which I presume is true. This purports the need to re-consider the operational definitions of risky behavior/practices based on evidence and standard definitions.

Response: We put this remark as possible justification because a study conducted at Addis Ababa (36.9%) was narrowly defined risky sexual practice as condom-unprotected sex with any partner and other characters like multiple sexual partners and casual sex were used as independent variables (not included in the risky behavior/practice) compared to our comprehensive definition. We believe this could be one possible reason. However, this justification cannot work compared to the result from Gondar because we used almost similar comprehensive operational definition. In this regard, revision of other possible reason (justification) was done. 

• Line 207-211= The discussion and final remark is not clear

Response: The comment was accepted and revision was made accordingly

• Line 212-219= The discussion lack clarity and coherence. The final remark is unrelated to issues mentioned in the preceding discussion and very stretched above and beyond the scope of the study. 

Response: This comment has been well taken and corrected accordingly

 

Reviewer 2:

• The background section outlines the research problem and talks about the issue of HIV and the problem in Africa, but three essential are missing: i) What hypothesis do the authors intend to test? ii) How does the article contribute to the literature? iii) How is the article innovative? I suggest the authors include those answers in the final background so that the readers will better understand the purpose of the manuscript.

Response: Response: Yes, this correct and we have accepted this comment 

• In line 120, the authors talk about the inclusion of variables with p <0.25. What is the reason for this value? The authors do not explain this. Are there any texts in the literature that use this value? If so, please include them in the manuscript. 

Response: We followed models building strategy steps mentioned by Hosmer and Lemeshow in applied logistic regression book. First, we used univariate analysis to identify important covariates. Secondly, all the variables with p-value < 0.25 with risky sexual practice in univariate analysis were fitted to the final multivariate logistic regression model. The comment was accepted and citation of the reference was made accordingly.

• In the results section, the authors should better explain Table 1 and the implication of the relative frequency on the sample on the results. As for income, I suggest adding a note explaining the values that represent poverty and extreme poverty.

Response: This comment has been well taken and corrected accordingly.

• In line 163, the authors write “About four fifth of the respondents had engaged in risky sexual practices”, but I did not find this information in the tables. I suggest that the authors include a table with risky sexual behaviors and the proportion of the sample with each behavior. I believe the table is more important than the text for the readers to understand the actual problem. Authors can explore Figure 1 further as well.

Response: Yes exactly, table is easier to understand, but it is difficult to present all findings by using a table. We use the text to reduce the number of tables in our manuscript. Correction was made by revising the text. 

• In Table 3, I suggest compiling the information about relationships in “lives with a partner or not”. You see, it is just a suggestion to turn the variables into just a binary. It is up to the authors to accept the suggestion or not. 

Response: length of stay in table 3 is to show the time duration of relationship among partners. Looking at other literature; we assumed this kind of classification could indicate the time interval that can affect risky practice. 

• Line 187, table note, has only one asterisk, for p <0.05. Authors should separate p <0.05 and p <0.01, as they represent different levels of significance.

Response: This is definitely correct and we have modified accordingly

---

## [Decision Letter · Decision Letter 1]

22 Sep 2020

PONE-D-20-13455R1

High level of risky sexual behavior among persons living with HIV in the urban setting of the highest HIV prevalence in Ethiopia: Implications for HIV interventions

PLOS ONE

Dear Dr. Tola,

Thank you for submitting your manuscript to PLOS ONE. After careful consideration, we feel that it has merit but does not fully meet PLOS ONE’s publication criteria as it currently stands. Therefore, we invite you to submit a revised version of the manuscript that addresses the points raised during the review process.

The manuscript still has many typos and grammatical errors. Please make sure that it is thoroughly edited by someone fluent in English. Please also correct the errors listed below.Please acknowledge that the sample size calculation was only made for estimating prevalence of risky sexual practice and not for estimating the factors associated with the outcome of interest. This should be discussed as a limitation under the discussion section.Line 137: It is not clear how the variable “condom use practice” is considered both as dependent and intendent variable.The data analysis section is superficial and does not tell how the independent variables were screened for the multivariable model.

We look forward to receiving your revised manuscript.

Kind regards,

Samson Gebremedhin, PhD

Academic Editor

PLOS ONE

Additional Editor Comments (Section-by-section comments):

Abstract

Line 21: Reconsider the use of the word “edge”

Line 26: “characters” >> “practices”

Line 28: “multivariate” >> “multivariable” (please do the same throughout the document)

Line 35: “were” >> “have”

Line 44 reconsider the use of the word “eradication”

Background

Line 49 “were” >> “had”

Line 60-62: The sentence is not clear

Line 74-75: what do you mean by “sexual exchange”? its not clear.

Line 85-87: please rephrase the sentence.

Line 91: Reconsider the use of the word “edge”

Methods

Line 96: “18 years and above” >> “18 years or above”

Line 98-100: Please remove the following sentence “The town is bordered by Gambella district and Bonga village by East, Oromia Regional State by Northeast, Gambella district by Northwest and West, and Abobo district by South.”

Line 109-111: remove the header “study population” and put the sentence “The study populations were all HIV positive adults attending ART from June to July 2019 at ART clinics in Gambella town.” as the first sentence of the “sample size and sampling procedure” header.

Line 115: please remove “CI”. Please note that confidence level and confidence interval are different.

Line 126 : “if” >> “when”

Line 127: Please mention the local language

Line 130: “characters” >> “practices”

Results

Table 1: please arrange all categories of the variables in decreasing order with the exception of those variables that have natural order (monthly income, family size)

Line 174: please reconsider the use of the word “were”

Table 2: what do you mean by “Caused intended”, please also indicate sample size for the variables.

Table 3: Bivariate >> Bivariable, Multivariate >> multivariable (please also do the same throughout the document)

Table 3: it is not clear how the candidate variables were identified for the multivariable model, This must be described in the data analysis sub-section

Table 3: Shisha/hashish use3x??? what do you mean by 3X?

Discussion

275-6: The sentence is not clear.

Line 284-85: Please remove the sentence “First, alcohol consumption in this study was not verified by using Alcohol Use Disorder Identification Test (AUDIT).”. It is not a must to use this tool.

Reviewers' comments:

Reviewer's Responses to Questions

**Comments to the Author**

1. If the authors have adequately addressed your comments raised in a previous round of review and you feel that this manuscript is now acceptable for publication, you may indicate that here to bypass the “Comments to the Author” section, enter your conflict of interest statement in the “Confidential to Editor” section, and submit your "Accept" recommendation.

Reviewer #2: All comments have been addressed

2. Is the manuscript technically sound, and do the data support the conclusions?

Reviewer #2: Yes

3. Has the statistical analysis been performed appropriately and rigorously? 

Reviewer #2: Yes

4. Have the authors made all data underlying the findings in their manuscript fully available?

Reviewer #2: (No Response)

5. Is the manuscript presented in an intelligible fashion and written in standard English?

Reviewer #2: Yes

6. Review Comments to the Author

Reviewer #2: (No Response)

7. PLOS authors have the option to publish the peer review history of their article (what does this mean?). If published, this will include your full peer review and any attached files.

Reviewer #2: No

---

## [Author Response · Author response to Decision Letter 1]

6 Nov 2020

Dear Editor:

I appreciate the comments and suggestions of the academic editor and reviewers; their comments and suggestions were constructive and have improved our manuscript substantially. Accordingly, I have incorporated almost all of the comments and suggestions given by the academic editor and reviewers in the manuscript. Furthermore, I have responded to each of the points raised by the reviewers as follows: 

Sincerely,

Tsegaye Berkessa (Corresponding author)

Academic Editor:

• The manuscript still has many typos and grammatical errors. Please make sure that it is thoroughly edited by someone fluent in English. Please also correct the errors listed below

Response: The comment was accepted and the manuscript was edited.

• Please acknowledge that the sample size calculation was only made for estimating prevalence of risky sexual practice and not for estimating the factors associated with the outcome of interest. This should be discussed as a limitation under the discussion section.

Response: This comment has been well taken and discussed in the limitation part. 

• Line 137: It is not clear how the variable “condom use practice” is considered both as dependent and intendent variable.

Response: The comment was accepted and all variables that measures outcome variable were removed from the list. 

• The data analysis section is superficial and does not tell how the independent variables were screened for the multivariable model.

Response: The comment was accepted and revision was made accordingly. We used model-building steps mentioned by Hosmer and Lemeshow in applied logistic regression book. All the variables with p-value < 0.25 with risky sexual practice in unavailable analysis were fitted to the final multivariable logistic regression model. 

Abstract

• Line 21: Reconsider the use of the word “edge”

Response: The comment was accepted and we replaced by another word. 

• Line 26: “characters” >> “practices”

Response: The comment was accepted and correction was made accordingly

• Line 28: “multivariate” >> “multivariable” (please do the same throughout the document)

Response: The comment was accepted and revision throughout the manuscript was made accordingly

• Line 35: “were” >> “have”

Response: The comment was accepted and correction was made accordingly

• Line 44 reconsider the use of the word “eradication”

Response: The comment was accepted and correction was made accordingly.

Background

• Line 49 “were” >> “had”

Response: This comment has been well taken and corrected accordingly.

• Line 60-62: The sentence is not clear

Response: The comment was accepted and the sentence was revised.

• Line 74-75: what do you mean by “sexual exchange”? its not clear.

Response: The comment was accepted and the phrase was clarified. 

• Line 85-87: please rephrase the sentence.

Response: The comment was accepted and the sentence was revised.

• Line 91: Reconsider the use of the word “edge”

Response: The comment was accepted and we replaced by another word.

Methods

• Line 96: “18 years and above” >> “18 years or above”

Response: Response: Correction was made accordingly

• Line 98-100: Please remove the following sentence “The town is bordered by Gambella district and Bonga village by East, Oromia Regional State by Northeast, Gambella district by Northwest and West, and Abobo district by South.”

Response: The comment was accepted and the sentence was removed from the manuscript. 

• Line 109-111: remove the header “study population” and put the sentence “The study populations were all HIV positive adults attending ART from June to July 2019 at ART clinics in Gambella town.” as the first sentence of the “sample size and sampling procedure” header.

Response: The comment was accepted and revised accordingly. 

• Line 115: please remove “CI”. Please note that confidence level and confidence interval are different.

Response: This is definitely correct and we have modified accordingly.

• Line 126 : “if” >> “when”

Correction was made accordingly

• Line 127: Please mention the local language

Response: The comment was accepted and local language was mentioned.

• Line 130: “characters” >> “practices”

Response: Revision was done accordingly

Results

• Table 1: please arrange all categories of the variables in decreasing order with the exception of those variables that have natural order (monthly income, family size)

Response: The comment was accepted and revised accordingly. 

• Line 174: please reconsider the use of the word “were”

Response: This comment has been well taken and corrected accordingly.

• Table 2: what do you mean by “Caused intended”, please also indicate sample size for the variables.

Response: “Caused intended” mean males who caused the pregnancy and reported that they intended to cause the pregnancy. This comment was accepted and this variable was removed from the manuscript due to sample size insufficiency, less than 10 events per variable (EPV).

• Table 3: Bivariate >> Bivariable, Multivariate >> multivariable (please also do the same throughout the document)

Response: The comment was accepted and revised accordingly.

• Table 3: it is not clear how the candidate variables were identified for the multivariable model, This must be described in the data analysis sub-section

Response: The comment was accepted and revised accordingly.

• Table 3: Shisha/hashish use3x??? what do you mean by 3X?

Response: It is editorial error. The comment was accepted and revised accordingly.

Discussion

• Line 275-6: The sentence is not clear.

Response: The comment was accepted and the sentence was revised.

• Line 284-85: Please remove the sentence “First, alcohol consumption in this study was not verified by using Alcohol Use Disorder Identification Test (AUDIT).”. It is not a must to use this tool

Response: The comment was accepted and the sentence was removed from the document.

---

## [Editor Report · Decision Letter 2]

9 Nov 2020

High level risky sexual behavior among persons living with HIV in the urban setting of the highest HIV Prevalent areas  in Ethiopia: Implications for interventions

PONE-D-20-13455R2

Dear Dr. Tola,

We’re pleased to inform you that your manuscript has been judged scientifically suitable for publication and will be formally accepted for publication once it meets all outstanding technical requirements.

Kind regards,

Samson Gebremedhin, PhD

Academic Editor

PLOS ONE
---

## [Editor Report · Acceptance letter]

13 Nov 2020

PONE-D-20-13455R2 

High level risky sexual behavior among persons living with HIV in the urban setting of the highest HIV Prevalent areas  in Ethiopia: Implications for interventions 

Dear Dr. Berkessa:

I'm pleased to inform you that your manuscript has been deemed suitable for publication in PLOS ONE. Congratulations! Your manuscript is now with our production department. 

Kind regards, 

on behalf of

Dr. Samson Gebremedhin 

Academic Editor

PLOS ONE